# RhoA Is a Crucial Regulator of Myoblast Fusion

**DOI:** 10.3390/cells12232673

**Published:** 2023-11-21

**Authors:** Chiara Noviello, Kassandra Kobon, Voahangy Randrianarison-Huetz, Pascal Maire, France Pietri-Rouxel, Sestina Falcone, Athanassia Sotiropoulos

**Affiliations:** 1Institut Cochin, Université de Paris, INSERM U1016, CNRS, F-75014 Paris, Francepascal.maire@inserm.fr (P.M.); athanassia.sotiropoulos@inserm.fr (A.S.); 2Centre de Recherche en Myologie, Sorbonne Université, INSERM UMRS 974, Institut de Myologie, F-75013 Paris, France; france.pietri-rouxel@upmc.fr

**Keywords:** adult muscle stem cells, muscle regeneration, small GTPase proteins, muscle damage, muscle hypertrophy, muscle cell fusion

## Abstract

Satellite cells (SCs) are adult muscle stem cells that are mobilized when muscle homeostasis is perturbed. Here we show that RhoA in SCs is indispensable to have correct muscle regeneration and hypertrophy. In particular, the absence of RhoA in SCs prevents a correct SC fusion both to other RhoA-deleted SCs (regeneration context) and to growing control myofibers (hypertrophy context). We demonstrated that RhoA is dispensable for SCs proliferation and differentiation; however, RhoA-deleted SCs have an inefficient movement even if their cytoskeleton assembly is not altered. Proliferative myoblast and differentiated myotubes without RhoA display a decreased expression of *Chordin*, suggesting a crosstalk between these genes for myoblast fusion regulation. These findings demonstrate the importance of RhoA in SC fusion regulation and its requirement to achieve an efficient skeletal muscle homeostasis restoration.

## 1. Introduction

Skeletal muscle is a highly plastic tissue, able to adapt to different events such as exercise or injury throughout life, which leads to a continuous recurrence of regeneration to maintain proper tissue homeostasis. Muscle regeneration is a fine-regulated process in which the main actors are the satellite cells (SCs), the adult muscle stem cells. They are located under the basal lamina in a quiescent state and, in response to injury, they exit this dormant state getting activated. This transition includes metabolic stimulation, cell-cycle entry, and migration [1,2,3]. Once dividing, only a subset of SCs self-renew to restore the quiescent pool, while the majority of them differentiate and fuse with each other in order to complete the formation of skeletal muscle fibers. For this reason, SC fusion is a required step during the regeneration process [4]. Although the transcriptional program governing skeletal muscle development and adult muscle myogenesis have been described in detail [5,6], the mechanisms that coordinate myoblast fusion remain partially understood. The conserved cell–cell fusion process so far described is composed of different steps: first, the recognition of two partner cells that have to migrate toward each other and bring themselves close enough (within 10 nm of one another) to allow the start of cells adhesion machinery; second, the induction of close membrane apposition through F-actin protrusions has to occur. The third step is the rearrangement of membrane lipid bilayer (hemifusion) and finally pore formation and cell cytoplasm fusion. In *Drosophila* it has been demonstrated that mechanical forces play an essential role in cell–cell adhesion and cell–cell fusion, and are essential to overcome repulsive force among myoblasts [7]. Due to hydration repulsion, the energy barrier is elevated and in this moment specialized proteins come in play; these proteins are called fusogenes and are required to initiate the fusion events [4,8]. In mammals, muscle-specific fusogenes, Myomaker and Myomerger, controlling different steps of the fusion process, hemifusion and pore formation respectively have been identified [9,10,11,12]. When Myomaker and Myomerger are co-expressed in fibroblasts, they are sufficient to induce fusion in these non-fusogenic cells [11]. Moreover, numerous proteins are necessary for all the steps involved in the complex process of cell fusion [13]. In particular, cell–cell adhesion molecules that are localized at the contact site such as M-cadherin, Integrin α9β1 and Adam12 [14] are important not only to enhance the contact between two cells but also to start signaling transduction pathway. In addition, remodeling of the actin cytoskeleton and actin dynamics are essential to achieve a correct fusion between myoblasts [15,16,17].

Using primary murine myoblasts, finger-like actin protrusions occurring at the site of fusion have been described for the first time by Randrianarison-Huetz and colleagues [18]. Characterized as essential structures for the fusion process, the molecular mechanism underlying their formation has been partially unveiled. In this paper serum response factor (Srf) has been identified as a master regulator of SC fusion required in both fusion partners. In particular, it is crucial to control the organization of the actin cytoskeleton and actin-based protrusions for myoblast fusion in order to achieve efficient hypertrophic myofiber growth [18]. Even if some downstream targets of Srf have been identified and could be implicated in the formation of actin-based protrusion regulating actin cytoskeleton organization, upstream regulators of Srf involved in this mechanism are still unknown. Interestingly Srf is one of the transcription factors activated by Ras homolog family member A (RhoA) [19,20] which translates physical forces into biochemical signalling and transmits this signal to the nucleus [21].

RhoA is a small GTPase protein that oscillates between GTP-bound and GDP-bound states, regulating a wide spectrum of cellular functions. Indeed, Rho GTPases coordinate the differentiation of several cell types and are involved in the regulation of immunological responses, blood pressure levels, and glucose homeostasis. Recent works revealed that Rho GTPases play a critical role during muscle development, regeneration, and function, suggesting that this protein is an important player in the modulation of both embryonic and adult myogenesis [22,23]. RhoA is also necessary for the initial induction of the myogenic program via the stimulation of SRF-mediated gene expression programs and, subsequently, to maintain the myoblasts in a proliferative state [24,25]. Later on, during the myogenesis process, the activity of this GTPase must be shut down to prompt cell cycle withdrawal and the final myoblast fusion step. Cell culture experiments have shown that RhoA fluctuates between high and low activity states in proliferating and differentiating myoblasts, respectively [26] underlying the complexity of its function. Thus, RhoA may control multiple aspects of muscle cell behavior. In this scenario, RhoA represents an interesting player to investigate during muscle adaptive responses. However, it remains to be determined whether such fluctuations also occur in vivo. Recently, it has been described that RhoA, activated in the whole muscle by Wnt4, maintains SCs in a quiescent state, promoting the inhibition of yes-associated protein (YAP) transcription factor in a Rho GTPase activates Rho kinase (ROCK) dependent manner [27]. Moreover, other works investigated RhoA in the modulation of the myogenic potential of cells evidencing its role in promoting myogenesis during muscle regeneration [28]. RhoA regulates muscle regeneration modulating autophagy flux and in consequence the switch from the quiescent to the activated state in satellite cells. Indeed, a recent study has shown that the inactivation of the RhoA–ROCK axis caused by the depletion of the upstream exchange factor ArhGEF3 promotes injury-induced muscle regeneration by increasing autophagy in mice [29]. Another work showed that Vav2, a Rho GTPase activator, modulates the signaling output of the IGF1- and insulin-stimulated phosphatidylinositol 3-kinase pathway in muscle tissue, including RhoA in modulator of muscle mass [30].

All these assertions show that Rho GTPases play important roles in muscle development, regeneration, and homeostasis. However, the redundant and/or compensatory effect of the other members of the Rho family represents a limitation in precisely defining the role of a single protein. In this paper we will focus on adult muscle stem cells. Moreover, the use of more sophisticated animal models (e.g., inducible, skeletal muscle-specific) may help to better define the role of Rho GTPases and identify their regulators and effector pathways in muscle homeostasis processes. Indeed, here we describe the role played by RhoA within SCs, taking advantage of conditional and inducible RhoA deletion, during skeletal muscle regeneration. We show that in the absence of RhoA, muscle regeneration and hypertrophy are compromised, mostly caused by impaired SC fusion. These defects are not correlated to actin disassembly or cytoskeleton shortfall. Furthermore, in differentiated primary muscle cells, we have evidence that RhoA ablation affects the expression of several genes, indicating novel functions of RhoA in adult skeletal muscle myogenesis and muscle plasticity.

## 2. Materials and Methods

### 2.1. Mouse Protocol

RhoA^lx/lx^ mice are homozygous for RhoA floxed alleles harboring LoxP sites flanking exon 3 of the endogenous RhoA gene [31]. Tg:Pax7-nGFP transgenic mice express nuclear localized EGFP under the Pax7 promoter [32].

To investigate the effect of satellite cell-specific RhoA-deletion in adult muscle, the mouse strain following mice were generated: Pax7-Cre-ER^T2^:RhoA^lx/lx^. In all experiments, 3-month-old Pax7-Cre-ER^T2^:RhoA^lx/lx^ mice (males and females) were given five intraperitoneal (i.p.) tamoxifen (Tam, 1 mg/day; MP Biomedicals, Santa Ana, CA, USA) injections to induce RhoA deletion (referred as Tam+). RhoA^lx/lx^ mice not injected with Tam were used as control mice (referred to as Tam−).

Mice were genotyped by PCR using the following primers: Cre-F 5′-CCTGGAAAATGCTTCTGTCCG-3′; Cre-R 5′-CAGGGTGTTATAAGCAATCCC-3′; RhoA^lx^-F 5′-AGGGTTTCTCTGTACGGTAGTC-3′; RhoA^lx^-R 5′-GCAGCTAGTCTAACCCACTACA-3′. All animal experiments were conducted in accordance with the European guidelines for the care and use of laboratory animals and were approved by the institutional ethics committee and the French Ministry of Research (number A751402).

### 2.2. Compensatory Hypertrophy (CH) Protocol

Compensatory Hypertrophy (CH) of *Plantaris* muscles of control and RhoA-deleted mice was induced through the incapacitation of soleus and gastrocnemius muscles by sectioning their tendon, in both legs. During the process of CH, both mice groups were injected with Tam on days 2 and 4 after CH. 3 weeks after CH, *Plantaris* muscles were dissected and processed for histological analyses. When indicated, mice were administered 25 µg/g EdU (Life Technologies).

### 2.3. Cardiotoxin Protocol

Muscle tissue injury in control and RhoA-deleted mice was achieved by a single intramuscular injection of 30 µL of 6 µM CTX (Latoxan, Valence, France) into *TA* muscle. During the process of regeneration, both mice groups were injected with Tam on days 3 after CTX. Mice were allowed to recover for 30 days and *TA* muscles were harvested.

### 2.4. Muscle Histology, Immunohistochemistry and Cell Staining

For assessment of tissue morphology, 8-μm-thick transverse sections were stained with hematoxylin and eosin (Hémalun de Mayer and Erythrosine 239 RAL Diagnostics) and examined under a light microscope.

*TA* muscles were collected and snap-frozen in liquid nitrogen-cooled isopentane. Eight μm-thick muscle sections were fixed in 4% PFA for 8 min at room temperature and blocked overnight at 4 °C in PBS 1×, 10% Horse serum, and 0.5% Triton X-100. Then, they were incubated with primary antibodies overnight at 4 °C in PBS 1×, 10% Horse serum, and 0.5% Triton X-100. The following primary antibodies were used: anti-eMHC (Alexis Biochemicals, San Diego, CA, USA, 805-504-L001, 1/50), anti-myogenin (sc-576 M-225, 1/100), anti-Laminin (Sigma, St. Louis, MO, USA, L9393, 1/200). After washes in PBS 1x, sections were incubated with secondary antibodies for 1 h at room temperature. The following secondary antibodies used were goat anti-mouse IgG1 Alexa 488 (ThermoFisher, Waltham, MA, USA, A21121, 1/1000) and donkey anti-rabbit Alexa 546 (Life Technologies, Carlsbad, CA, USA, A10040, 1/1000). Nuclei staining was performed using DAPI (Sigma, St. Louis, MO, USA, 14530, 1/10000). Muscles sections were then mounted in Dako Fluorescence Mounting Medium and kept at 4 °C until image acquisition.

For Pax7 staining, muscle sections were fixed in 4% PFA for 8 min at RT and permeabilized in ice-cold methanol for 6 min. Muscle sections were treated with Antigen Unmasking Solution pH6 (Vector, H-3300) for 15 min at 95 °C and cooled on ice for 30 min. Blocking and incubation with primary and secondary antibodies were conducted as described in the previous paragraph. Primary mouse anti-Pax7 antibody (Santa Cruz, sc-81648) was used at dilution 1/50.

EdU detection was performed using the Click-iT EdU Alexa Fluor 647 kit, according to the manufacturer’s instructions (Life Technologies).

Apoptotic cell detection was performed using ApopTag^®^ Red Apoptosis Detection Kit (S7165), according to the manufacturer’s instructions (EMD Millipore, Chicago, IL, USA).

Muscle cells cultured in dishes were fixed for 8 min in 4% PFA and then permeabilized and blocked in PBS with 0.1% Triton X-100 and 5% horse serum for 1 h at RT. Cells were incubated overnight at 4 °C with the following primary antibodies: anti-MHC (DSHB,1/50) and anti-pMLC2 (cell signaling, 3671S, 1/1000) diluted in the same buffer. After incubation for 1 h at RT with fluorescent secondary antibody anti-mouse IgG1 Alexa 488 (1/1000; A21121), cells were stained with DAPI (for nuclei) and phalloidin Alexa Fluor 488 (1/500; Thermo Fisher Scientific; for F-actin) and mounted in Fluorescent Mounting Medium (Dako).

### 2.5. Primary Muscle Cell Culture

Primary cultures were derived from hindlimb muscles of control and RhoA-deleted of 6- to 8-week-old mice all harboring the Pax7-nGFP transgene that allowed prospective selection of SCs by FACS. The dissection of the muscles was performed with care to take off as much fat and connective tissue as possible. The muscles were minced in DMEM/F12 supplemented with 2% antibiotic/antimycotic (15240-062; Gibco, Waltham, MA, USA) in a sterile Petri dish on ice. The minced muscles were digested three times for 25 min at 37 °C with 1 mg/mL collagenase D (Roche) and 0.1% Trypsin (15090-046; Gibco, Waltham, MA, USA), and digestion was stopped by adding FCS (25% final). Cells were filtered through a 70-µm cell strainer and pelleted. Cells were then washed three times in DMEM/F12 and 2% antibiotic/antimycotic, resuspended in 1× PBS without Ca^2+^ and Mg^2+^, 2% FCS, and 2% antibiotic/antimycotic, and finally filtered with a 40-µm cell strainer. Pax7/GFP-positive SCs were sorted on FACS Aria III (BD) previously calibrated (fluorescence minus one and use of compensation beads) using the CYBIO Cochin Institute platform. Cells were collected in a FACS tube containing FCS and 2% antibiotic/antimycotic.

In standard conditions, myoblasts were grown in a growth medium (DMEM/F12, 2% Ultroser G [PALL Life Sciences, Portsmouth, United Kingdom], and 20% FCS) on plastic dishes coated with 0.02% Gelatin. For differentiation, myoblasts were seeded in Matrigel-coated dishes and cultured in a differentiation medium (DMEM/F12 and 2% horse serum).

### 2.6. Proliferation Assays

To detect S-phase entry, control and RhoA-deleted SCs were plated immediately after sorting, cultured for 5 days in a growth medium, and pulsed with EdU (10 µM; Life Technologies, Carlsbad, CA, USA) for 2 h before fixation with 4% PFA. EdU detection was performed using a Click-iT EdU Alexa Fluor 647 kit, according to the manufacturer’s instructions (Life Technologies, Carlsbad, CA, USA).

### 2.7. RNA Extraction and RT-qPCR

Total RNA was extracted using TRIzol reagent and reverse-transcribed with SuperScript III reverse transcriptase (Invitrogen). cDNA was synthesized from 1μg of RNA. Quantitative PCR analysis was performed using a Light Cycler (Roche) according to the manufacturer’s instructions using a SYBR Green I kit (Roche). Values were normalized using *Hydroxymethylbilane synthetase* (*Hmbs*). The following primers were used: RhoA-F 5′-AACCTGTGTGTTTTCAGCACC-3′; RhoA-R 5′-ACCTCTGGGAACTGGTCCTT-3′; Hmbs-F 5′-TGCACGATCCTGAAACTCTG-3′; Hmbs-R 5′-TGCATGCTATCTGAGCCATC-3′; MYH3-F 5′-GCAAAGACCCGTGACTTCACCTCTAG-3′; MYH3-R 5′-GCATGTGGAAAAGTGATACGTGG-3′; Myogenin-F 5′-GAAAGTGAATGAGGCCTTCG-3′; Myogenin-R 5′-ACGATGGACGTAAGGGAGTG-3′; MyoD-F 5′-GGCTCTCTCTGCTCCTTTGA-3′; MyoD-R 5′-AGTAGGGAAGTGTGCGTGCT-3′.

### 2.8. Cell Counting Analysis

To determine the average number of nuclei per fiber, we quantified the total number of DAPI+ nuclei contained within each muscle fiber or myotube.

The fusion index was calculated as the fraction of nuclei contained within MyHC^+^ myotubes which had two or more nuclei, as compared to the number of total nuclei within differentiated cells (expressing MyHC).

In order to specify if fused cells are binucleated or bigger myotubes we determined the ratio between the number of nuclei contained within all MyHC^+^ cells and the number of cells MyHC^+^.

The differentiation index was calculated as the fraction of nuclei contained within all MyHC^+^ cells, including both mononuclear and multinuclear cells, as compared with the number of total nuclei within each image. Cells were considered positive for MyHC staining when the fluorescence intensity was clearly above background levels.

### 2.9. Cell Migration Assay

The migration of primary mouse muscle cells was quantified using time-lapse microscopy. Myoblasts were seeded in gelatin-coated eight-well Ibidi plates and maintained in a rich medium. The next day, cells were filmed using an inverted Axio Observer Z1 microscope (Zeiss) with an LCI PlN 20×/0.8 W DICII objective and an incubation chamber at 37 °C and 5% CO_2_. Live cells were monitored every 6 min for 4 h and 30 min with bright-field and Metamorph 7.7.5 software. Cell velocities were calculated in µm per minute using ImageJ (National Institutes of Health) by tracking the paths of cells. At least 100 cells were tracked for each group of two independent cell cultures.

### 2.10. Western Blot Analysis

Myoblast control and mutant were lysed in RIPA buffer (Sigma, St. Louis, MO, USA) and proteins were separated through denaturation SDS-PAGE electrophoresis using Mini-Protean TGX precast gels 4–15% (Biorad, Hercules, California, USA) and transferred on Nitrocellulose (0.2 micron, Biorad) membrane using the Trans-Blot turbo transfer system (Biorad). Membrane were blocked with 5% skinned milk in TBS-1% Tween (TBST) 1 h at room temperature and probed overnight at 4 °C with primary antibody. The following antibody were used: rabbit anti pMLC2 (1/50; #3674; Cell Signaling) and mouse anti α Tubulin (1/4000; T6064; Sigma, St. Louis, MO, USA)

Following washing in TBST, membranes were hybridized with secondary antibodies goat anti-mouse coupled to HRP (ThermoFisher, Waltham, MA, USA, 62-6520). Proteins were revealed using SuperSignal West Femto substrate (ThermoFisher, Waltham, MA, USA).

### 2.11. Image Acquisition

Digital images were acquired using an Olympus BX63F microscope with 10× objective (UplanFL, numerical aperture 0.3) and 20× objective (UPLSAPO, 0.75), ORCA-Flash4.0 LT C11440-42U camera (Hamamatsu); an Axiovert 200 M microscope (Zeiss) with 5× objective (PLANFLUAR, 0.25) and 20× objective (LD PLANNEOFLUAR, 0.4), cooled CCD CoolSNAP-HQ2 camera (Photometrics); or a Spinning Disk Leica confocal microscope with a 100× oil-immersion objective (HCX PL APO, 1.47), cooled CCD CoolSNAP-HQ2 camera (Photometrics) and Metamorph v.7.7.5 (Molecular Devices). Images were composed and edited in ImageJ. The background was reduced using brightness and contrast adjustments applied to the whole image.

### 2.12. Affymetrix Microarrays

Microarray analysis was performed from three independent control and mutant cell cultures. Total RNAs were obtained from cells at day 0 (corresponding to myoblasts), day 1 (corresponding to myocytes), and day 3 (corresponding to myotubes) of differentiation, using RNeasy Mini kit (Qiagen, Les Ulis, France) and DNase treatment (Qiagen, Les Ulis, France). RNA integrity was certified on a bioanalyzer (Agilent, Santa Clara, CA, USA). Hybridization to Mouse Gene 2.0-ST arrays (Affymetrix) and scans (GCS3000 7G Expression Console software V1.4) were performed on the Genom’ic platform (Institut Cochin, Paris, France). Probe data normalization and gene expression levels were processed using the robust multiarray average (RMA) algorithm in the expression Console (Affymetrix, Santa Clara, CA, USA). Gene ontology analysis was performed using Ingenuity (IPA) software, version 51963813 (Release Date: 11 March 2020). Full data are available on Gene Expression Omnibus: GSE242637.

### 2.13. Morphometric Analysis

The Myofiber cross-section area (CSA) was analyzed by using immunostaining of Laminin, marking myofiber sarcolemma, and then using the MuscleJ tool [33] or ImageJ macro previously developed in our laboratory [18]. Between 1300 and 3000 myofibers were analyzed in regenerative areas. For the quantification of the number of nuclei per myofibers, ImageJ was used and at least 1300 myofibers were counted per muscle.

## 3. Results

### 3.1. The Presence of RhoA in Satellite Cells Is Necessary for a Correct Muscle Regeneration

To evaluate the role of RhoA in adult SCs, we studied the effect of SC-specific *RhoA* deletion on skeletal muscle homeostasis and regeneration. In particular, we used Pax7^CreErt2/+^:RhoA^lx/lx^:Pax7-nGFP two-month-old mice strain, injected or not with tamoxifen (Tam). Tam-injected mice present SC-specific *RhoA* deletion, referred to as mutants (Mut) and they are compared to non-injected mice, referred to as controls (Ctl). Efficient loss of RhoA was validated at the transcript level on Fluorescence Activated Cell Sorting (FACS) sorted SCs (Pax7:nGFP) from control and RhoA deleted mouse muscles (Figure 1A).

At first, we compared Ctl and Mut at basal state. We observed that the absence of RhoA did not affect the body weight of animals (Figure 1B).

Then, we investigated the number of SCs on the *Tibialis Anterior* (*TA*) muscle section and we observed there was no evident difference between Ctl and Mut groups of mice in terms of the number of Pax7^+^ cells, a specific marker of these cells [34] (Figure 1C). Moreover, we counted by FACS the number of SCs purified by Pax7-nGFP transgene expression and we confirmed that at the basal state, the absence of RhoA did not affect the survival of SCs (Figure 1D). Related to a previous paper that showed the importance of RhoA to maintain SCs in quiescence [27,35], in addition to SC number, we assessed their niching by counting the SCs located outside the basal lamina. Our data showed that the majority of SCs lacking RhoA were well located, under the basal lamina (Figure 1E), indicating that the absence of RhoA does not induce an impaired SC localization.

To activate SC-mediated muscle repair, we induced injury by injecting Cardiotoxin (CTX) into *TA* muscles (Figure 2A). We then analyzed the efficacy of regeneration at different time points by harvesting *TAs* at 4, 8, 14, and 30 days after CTX injury (Figure 2B). The regeneration process of SC-RhoA-deleted muscle was impaired and 14 and 30 days after CTX injury *TA* muscle masses of SC-RhoA-deleted mice were smaller (Figure 2C).

Once we made sure that body weight was not affected by the absence of RhoA during the regeneration process (Appendix A), we wanted to determine whether the decreased muscle mass in SC-RhoA-deleted mice resulted from the reduced myofibers size or from reduced myofiber number, so we quantified both parameters. The absolute number of myofibers per muscle counted at different time points during the regeneration process did not vary between the Ctl and Mut groups (Figure 2D). In contrast, the cross-section area (CSA) of regenerated myofibers was strongly diminished in muscle lacking RhoA in SCs (Figure 2E).

### 3.2. The Number of Satellite Cells Lacking RhoA Is Transiently Altered Causing a Delay in the Regeneration

As the regeneration of mature myofibers is dependent on SCs, we investigated the behavior of SC-RhoA-deleted (Mut) as compared to controls (Ctl).

We first quantified the absolute number of SCs on *TA* muscle sections based on immunostaining of Pax7, comparing muscles with the same regeneration area (more than 60%). As we showed in Figure 1C at basal state, with no CTX injury, there is no effect of RhoA absence. While at four and eight days after injury, SCs lacking RhoA were significantly less than control, such difference is not observed at days 14 and 30 (Figure 3A). To further validate these results obtained using immunostaining/light microscopy, we counted by FACS the number of SCs purified by Pax7-nGFP transgene expression three days after CTX injury. We found that the SC number was significantly less in SC-RhoA-deleted muscles (Figure 3B), confirming the quantification obtained by imaging.

Histological analysis of hematoxylin/eosin stained muscle sections revealed that four days after CTX injection, myofibers were not evidenced (Figure 3C), suggesting the presence of ghost fibers, surrounded by basal lamina [36]. To identify the newly formed myofibers we immunostained the regenerating muscles for embryonic Myosin Heavy Chain (eMHC), being expressed during embryonic development and transiently in adult muscle fibers during regeneration [37]. We found that eMHC had a different expression pattern in the SC-RhoA-deleted muscles compared to the control four days post-injury (Figure 3C), indicating that an altered regeneration process was occurring. The quantification of eMHC^+^ myofibers four days after muscle injury evidenced a significant reduction of regenerating fibers (Figure 3D). Moreover, the CSA of newly formed eMHC-positive myofibers was smaller in the absence of RhoA compared to the control regenerated muscles suggesting a delay in the regeneration process in the mutant (Figure 3E). These data show that RhoA in SCs is necessary to have a correct regeneration process and myofiber maturation.

### 3.3. RhoA Is Dispensable for SC Proliferation

The transient decrease of SC number could be due to an imbalance between cell apoptosis and/or proliferation or a problem in their activation upon injury. In order to understand if the reduced number of Pax7^+^ cells observed in the RhoA-deleted muscles at an early stage of regeneration could be attributed to altered cell survival, we performed a TUNEL assay on muscle sections four days after CTX injury. We did not observe double stained cells for Pax7 and TUNEL, indicating that both SC-control and SC-RhoA-deleted did not undergo increased apoptotic cell death (Appendix A).

We next wondered if a reduced number of SCs might come from an alteration of cell proliferation in the RhoA-depleted condition. To investigate this parameter in vivo upon CTX-injury, we injected intraperitoneally EdU, 24 h and 5 h before mouse euthanasia. EdU is a nucleoside analog of thymidine incorporated into DNA during its active synthesis, staining the S-phase entry of the cells. By double staining EdU and Pax7 on *TA* muscle cryosections, we asserted the percentage of proliferating satellite cells and revealed no difference in S phase-SCs between SC-RhoA-deleted and SC-control. Nevertheless, we observed an extensive SC proliferation four days after CTX that progressively came back to quiescence at day 30 (Figure 4A).

To understand if this result depended intrinsically on SCs, EdU incorporation was assessed in vitro, in proliferating FACS-sorted myoblasts from control and SC-RhoA depleted muscles. After five days in culture, cells received a pulse of EdU for 2 h. We assessed that the percentages of EdU+ cells of MB-control and RhoA-deleted were equivalent, indicating similar proliferation rates between control and RhoA-deleted myoblasts (Figure 4B).

All these data suggest that the absence of RhoA does not have any impact on SC survival and does not affect their proliferation in vivo and in vitro, advancing the option that maybe there is an issue in their activation and their first entry into the cell cycle.

### 3.4. RhoA Is Not Required for Myoblast Differentiation

To further investigate the potential role of RhoA in SC behavior during the regeneration process, we assessed the differentiation potential of control and RhoA-deleted SCs in vivo. For this purpose, we verified the expression of the canonical myogenic differentiation marker *Myogenin*, three days and eight days after CTX injection, two crucial time points during muscle regeneration [38]. *Myogenin* expression levels at these two time points post-injury did not differ between control and RhoA-deleted cells (Appendix A). Moreover, when we quantified Myogenin (MyoG) protein expression by immunofluorescence on cryosections fourteen days after CTX injury, no difference in MyoG+ nuclei was evidenced between control and RhoA-deleted muscles (Figure 5A,B).

To gain further insights, differentiation potential was investigated in vitro on primary sorted control and RhoA-deleted SCs. In line with in vivo data, *MyoD* and *Myogenin* transcript levels did not differ between the two conditions under proliferation (day 0) and differentiation conditions (day 1 and day 3) (Figure 5C). In addition, we monitored the expression of late differentiation marker (Myosin Heavy Chain, MyHC) in control and RhoA-deleted cells at day three post differentiation using MF20 antibody. Quantification of differentiation index (i.e., the number of nuclei in MyHC^+^ cells normalized on the number of total nuclei) revealed that the absence of RhoA did not impair further stages of myogenic differentiation as both cell groups exhibited a similar differentiation index (Figure 5D,E). On the contrary, RhoA absence enhanced the amount of nuclei expressing MyHC. Altogether, these data showed that RhoA is not essential for SC differentiation.

### 3.5. RhoA Is Necessary for Primary Myoblast Fusion

We thus investigated the fusion capacity of SCs, crucial to allow the formation of newly regenerated myofibers.

To assess this parameter, the number of sub-sarcolemmal myonuclei was quantified in RhoA-depleted and control *TA* muscle sections, at different time points after CTX injection. In mutant muscles, the number of myonuclei per myofibers was significantly decreased at each time point after muscle injury compared to control muscles (Figure 6A). Moreover, there is a growth of the muscle during regeneration as shown by the increase in myonuclei number between 14 and 30 days after CTX injection, indicating a continuum of SC fusion to the regenerated recently formed myofibers, in the mutant muscles the number of myonuclei per myofibers remains constant over all the regeneration period (Figure 6A). These data suggest that the lack of RhoA in SCs elicits an important fusion defect and, consequently, impaired muscle regeneration and muscle growth.

To validate this result, the fusion ability of control or RhoA-deleted SCs was investigated also in muscles submitted to increase workload (overload condition). Indeed, in this situation compensatory hypertrophy (CH) takes place, requiring SC fusion as an important event allowing the growth of myofibers [39,40]. During muscle regeneration, the fusion occurs at first between SCs and in later stages between SCs and regenerated myofibers in order to achieve muscle growth. In that case, both fusion partners (satellite cells or regenerated myofibers) are RhoA-deleted. In contrast, during hypertrophy induced by overload, fusion occurs between activated SCs and growing myofibers that in our model diverge in RhoA expression (RhoA expressing myofibers and RhoA-lacking SCs). We found that the absence of RhoA in SCs affected the hypertrophic growth of *Plantaris* muscle (Figure 6B). To further analyze this alteration, we assessed the fusion capacities of SCs in mice subjected to CH by chronically injecting EdU (from the third to the eleventh day after surgery), to track EdU+ nuclei incorporated into growing myofibers. As expected based on *TA* muscle data shown in Figure 1C, no difference was evidenced between Ctl and Mut *Plantaris* muscles in Sham Operated condition. However, three weeks after overload, the percentage of EdU+ myofibers was significantly reduced in RhoA-deleted muscles compared to control muscles (Figure 6C). This defect evidences an impaired fusion of RhoA-lacking SCs into wild-type growing myofibers. The fact that the fusion event was strongly affected by the absence of RhoA in SCs and that RhoA in myofiber is important to have correct muscle growth [40], suggests that RhoA is necessary in both SCs and myofibers to regulate fusion and muscle growth.

To better establish the fusion potential of RhoA-depleted SCs, we performed in vitro experiments on primary myoblasts. FACS-sorted SCs were collected from mutant or control mice and differentiated in culture for three days. SC-RhoA-deleted were unable to form proper myotubes (Figure 5D and Figure 6D) and the number of myonuclei per myotubes was much lower in the RhoA-deleted cells compared to the control (Figure 6E). Accordingly, the fusion index, which represents the proportion of nuclei within multinucleated cells (with at least 2 nuclei) among MyHC^+^ cells normalized on the total nuclei number, was lower in RhoA-depleted myotubes compared to the control (Figure 6F).

Collectively, these data showed that RhoA loss in SCs strongly affects myogenic fusion events. Moreover, the fact that both the fusion index and the number of myonuclei per myotube/myofiber are decreased suggests that both fusion SC/SC (primary fusion) and SC/growing myofiber (secondary fusion) are impaired.

### 3.6. RhoA Is Required for a Correct and Efficient Migration of SCs

As coordination of migration and fusion of SCs is essential for the correct regeneration process [41], we next wondered whether RhoA could also play a role in myoblast migration thus contributing to the altered regeneration phenotype of RhoA deleted muscles. Cell migration was monitored by live cell imaging and velocity was measured. Mean velocity was calculated for individual myoblasts (control and RhoA-deleted SCs) (Figure 7A). We did not find a significant difference in migration among RhoA-deleted and control cells. However, RhoA, together with other members of the Rac family (Cdc42 and Rac), has been described as involved in lamellipodia and filopodia formation that determines the direction of the migration at the cell’s edge [42]. Considering this property, we analyzed the traces of the migration path of RhoA-deleted and control myoblasts and we found that the migratory path of control SCs was linear with an elevated distance between the point of origin and the point of the end of the record, contrary to the RhoA-deleted cells where the path was shorter and contorted (Figure 7B). All these data indicated that RhoA plays a role in a correct and efficient migration even though velocity is not affected.

### 3.7. Cytoskeleton Integrity Is Not Affected by the Absence of RhoA

RhoA has been reported as a major regulator of cell contractility and actin cytoskeletal reorganization, inducing the formation of stress fibers and focal adhesions [43]. Thus, we investigated whether RhoA-deleted myoblasts present cytoskeleton defects that could underlie the fusion deficiency. Confocal microscopy analysis of control and RhoA-deleted proliferating myoblasts stained for F-actin did not reveal recognizable differences, and actin cables and cortical actin were clearly distinguishable in both RhoA-deleted and control cells, with no difference in phalloidin staining intensity progressing toward the top of the cells on confocal ȥ-sections (Figure 7C).

ROCK is one of the major downstream targets of RhoA and its substrate myosin-binding subunit (MBS) leads to an increase in Myosin Light Chain 2 (MLC2) phosphorylation and consequently induction of actomyosin contraction [44]. We wondered if the decreased fusion capacity of RhoA lacking myoblasts could be due to a defect in cell contractility originating from impairment of MLC2 phosphorylation, as expected after perturbation of the RhoA–ROCK axis. Indeed, Myosin II protein and cortical rigidity have been described to mediate the invasion of fusion partners and its mutation leads to myoblast fusion impairment in *Drosophila* [7]. Elsewhere it was shown that Blebbistatin (a potent inhibitor of myosin II ATPase activity) induces an inhibition of fusion, suggesting a potential role for actomyosin bridging and Ca^2+^ channels in the fusion process and that myosin II motor activity is essential for myoblast fusion [16]. In our hands, analysis of the total amount of pMLC2 signal, quantified by Western blot, revealed that there is no change between RhoA-deleted and control cells, suggesting that one of the causes of fusion defects in RhoA-lacking myoblast is not pMLC impairment (Figure 7D).

### 3.8. RhoA-Deleted Muscle Cells Display an Impaired Gene Expression

To identify putative genes and signaling pathways involved in the impaired fusion of RhoA-deleted myoblasts, we performed a microarray analysis on myoblast expressing or not RhoA at three stages: proliferating myoblasts (day 0), differentiation-committed myocytes (day 1) and differentiated myotubes (day 3). Only 17 genes displayed an altered expression in all conditions following the loss of RhoA (Figure 7E). The most downregulated gene that emerged on day 0, day 1, and day 3 of the differentiation process, was the *Chordin* (Appendix A). We validated by RT-qPCR its downregulation on day 3 (Figure 7F), suggesting a role of this organizer-specific secreted protein and a BMP antagonist on the impaired fusion observed in RhoA-deleted myoblasts. Additionally, we found 980 genes differently expressed between control and RhoA-deleted cells on day 0, 423 on day 1, and 614 on day 3, showing that the genes differentially expressed do not overlap between the three stages. These data suggest that the functions orchestrated by RhoA are different depending on the specific cell feature and cellular context and that RhoA might act on different signaling pathways in these different cell stages (myoblasts, myocytes, and myotubes). This hypothesis was corroborated by the fact that among the Canonical Pathway predicted by the gene ontology program as differently regulated between control and RhoA-deleted cells, there was very little overlap between day 0, day 1, and day 3 (Appendix A).

Focusing our attention on the upstream regulators predicted and activated (Z-score > 2) or inhibited (Z-score < 2) in RhoA-deleted cells compared to the control using IPA analysis (*p*-value < 0.05) (Appendix A), we noticed that at day 3 of differentiation, among the upstream regulator significantly predicted inhibited, there was Adam12, a disintegrin and metalloprotease identified as a potent stimulator of myoblast fusion [45]. Interestingly, it was shown that Adam12 and α9β1 Integrin mediate a cell–cell interaction selectively involved in the fusion of mononucleated myoblasts to preformed myotubes [14]. Among the target molecules associated in the dataset with the Adam12 upstream regulator, there are some extracellular matrix protein genes, such as the proteoglycan (collagen-associated) Decorin (*Dnc*) and Dermapontin (*Dpt*). To validate these data, we quantified by RT-qPCR the expression changes of some of the most relevant genes modulated in mutant differentiated myotubes used in the transcriptomic analysis. We confirmed that only the expression of *Dcn* was increased in the absence of RhoA on day 3 (Appendix A), while *Dpt* expression did not result in increased RT-qPCR quantification. These data suggest a possible involvement of Dcn in myoblast fusion defects observed in the absence of RhoA. Altogether, these expression studies show that, in muscle cells, RhoA affects various pathways depending on the differentiation state, suggesting that different functions of RhoA might be implicated in RhoA-deleted myoblast fusion defects.

## 4. Discussion

Skeletal muscle regeneration and muscle hypertrophy rely on sequences of fine-correlated events, crucial for complete and functional muscle restoration, formation, and growth. In this study, we aimed to establish the role of RhoA, expressed specifically in SCs in this physiological process by using an inducible SC-targeted knock-down mouse model. We demonstrated that RhoA deletion in SCs negatively affected skeletal muscle regeneration and growth. RhoA-deleted SCs are less in number compared to control SCs after muscle injury, suggesting trouble in their activation, although their proliferation and their differentiation were not affected either in vivo or in vitro. Importantly we showed that RhoA is indispensable for satellite cell fusion both in vitro and in vivo upon regeneration and hypertrophy in a cell-autonomous manner by affecting the expression of genes not yet described as important for this function in mammals.

RhoA has been described as an important regulator of cell cycle progression of some human cancers and cancer-associated mutations in Rho family regulators have been characterized [46]. In contrast to these previous results, we demonstrated that the proliferation of RhoA-deleted SCs was not affected, in vivo and in vitro, excluding that RhoA could regulate muscle cell proliferation.

Despite a comparable number of RhoA-deleted SCs vs. control conditions, we found a reduced number of these cells in the early days of the regenerative process, suggesting an impairment in their ability to be activated. Nevertheless, Eliazer et al. showed that RhoA activation reinforces SC quiescence and when it was disrupted in SCs induces an abnormal activation of SCs in uninjured muscle [27]. Our study did not confirm these data maybe because the mouse models used are different. In both cases, the deletion occurs specifically in the satellite cells but in the present study mice used are RhoA^lx/lx^ and the mice used by Eliazer et al. are RhoA^lx/+^. In this later study, the authors reported a mild reduction of RhoA activity in SCs, but there is no report on residual RhoA expression. On the contrary in our case, we have a complete (almost 100%) reduction of RhoA expression.

At present, we cannot exclude the differences in the methods used to analyze SC proliferation. Moreover, this paper underlines the involvement of RhoA in cytoskeleton-related signaling (like YAP and actin) and whether we support the idea (based on array data) that RhoA could have alternative possible targets regulated in the regeneration context.

However, we cannot exclude a compensatory effect of other members of the Rho family in our model. Moreover, contrary to our results, they observed that after reduction in RhoA, SCs are located outside the niche, a finding that we did not observe in our model, indicating that the impaired muscle regeneration was independent of altered niching.

Anyway, further analyses are essential to elucidate the role of RhoA in SC activation.

We demonstrated that the differentiation of RhoA-deleted SCs was not affected in vivo or in vitro. This is in agreement with published data on the C2C12 cell line showing that both constitutive RhoA activation (RhoAV14) and RhoA activity inhibition (C3 transferase treatment) did not affect differentiation [26]. It has been shown that RhoA activity must be tightly regulated in a finely coordinated time-dependent manner to ensure appropriate skeletal muscle formation. Indeed, in mouse C2C12 myoblast RhoA activity has been found to decrease in a biphasic manner during myogenic differentiation [47]. Here, we showed for the first time in primary murine myoblasts that the complete deletion of RhoA does not affect differentiation. Moreover, mutant myotubes have a slightly increased differentiation index underlying the concept that the RhoA function should be finely modulated during all the steps to have a correct myogenesis.

RhoA has been described as an important regulator of actin polymerization and cytoskeleton assembly [42]. However, in our hands, RhoA-deleted myoblasts did not present evident defects in F-actin cytoskeleton organization. Moreover, the formation of F-actin bundles did not appear affected by the absence of RhoA. However, we clearly showed that RhoA affected cell motility: not cell velocity but cell direction. Recent work demonstrated that syndecan-4 regulates the correct polarization of migrating mammalian myoblasts, in part in coordination with RhoA [48,49]. Our work and this last cited paper provide the first foundations to further elucidate the role of RhoA and its partners in SC motility orchestration.

In this study, we demonstrated that the main step impacted by the absence of RhoA is SC fusion. By CTX-induced regeneration we showed that primary (myoblast-myoblast) and secondary (myoblast-myotube) fusions were affected in RhoA-depleted SCs, considering the fact that the number of myonuclei per myofibers is reduced during all the regeneration process, impacting myofibers growth. This concept is also confirmed in vitro where not only the number of myonuclei per myofibers was reduced but also the fusion index, meaning that there is an impairment in the fusion of two mononucleated cells and of myotube and mononucleated cells. Moreover, during muscle overload-induced CH, we showed that the fusion of RhoA-deleted SCs to the growing myofibers is impaired as well, suggesting that even in heterotypic milieu (myofiber expressing RhoA/SC RhoA deleted), RhoA-deleted SCs are not able to fuse. These experiments suggest that the absence of RhoA in SCs renders them incapable of fusing either to myoblasts (regeneration situation) or to myofibers, RhoA-deleted (during muscle regeneration) or RhoA-expressing (during muscle hypertrophy).

Indeed, despite the reduced number of myonuclei per fiber 4 days after muscle damage, the number of satellite cells 14 and 30 days after ctx injury is comparable between control and mutant, suggesting that the decreased amount of myonuclei per myofiber might principally depend on fusion defect occurring in SCs lacking RhoA per se.

Furthermore, we showed that this defect was cell autonomous as murine primary SCs lacking RhoA were unable to fuse in vitro. Reports already published on C2C12 cells have asserted that RhoA must be deactivated to enable myoblast fusion [50]. Somehow, our results confirm that RhoA expression/activity should be carefully balanced during myogenesis because of the deleterious effects of its chronic deletion on SC fusion.

Surprisingly, the absence of RhoA in proliferative myoblasts and differentiated myotubes did not affect the expression of genes already known to be involved in muscle cell fusion. Indeed, there was no difference in the expression of *Myomaker* or *Myomixer* in differentiated RhoA-deleted cells. Even the expression of *Srf*, a master regulator of myoblast fusion, was not affected in RhoA-deleted cells, explaining in part why there is no actin assembly issue [18]. Indeed there is no overlap with the genes deregulated in Srf deleted muscle cells [18] suggesting that the impaired fusion occurring in the RhoA deleted model is independent of the cytoskeleton issue and of the RhoA/Srf axis. We can suppose that RhoA in SCs interacts with new partners not yet identified and that there are other RhoA downstream effectors controlling SCs fusion and muscle growth.

One gene we found downregulated on day 0, day 1, and day 3 of the differentiation process of RhoA deleted cells is the *Chordin*. Chordin is an important negative regulator of BMP activity by inhibiting the binding of these ligands to their receptors [51]. Previously, it has been shown that *Chordin* expression increases during the differentiation process in C2C12 cells. This is in line with the fact that Chordin, as an intrinsic inhibitor of BMP signaling, supports myoblast differentiation and fusion [52]. In our hands, *Chordin* expression decreased in the absence of RhoA, suggesting that its role in BMP pathway inhibition was reduced and, in consequence, the fusion program was altered.

In our previous work, we described that among the most downregulated genes by the absence of RhoA in myofibers there were genes associated with extracellular matrix rearrangement [40]. Noteworthy, among the Upstream Regulators significantly predicted inhibited (Z-score < 2) in RhoA-deleted three days differentiated myotubes, there was Adam12, a disintegrin and metalloprotease protein. We can speculate that Adam12, as an upstream regulator, controls matrix protein regulating adhesion and fusion of primary myoblast.

In conclusion, we described a novel role for RhoA within SCs during skeletal muscle regeneration. We propose that RhoA mainly affects SC fusion in part through a modification of SC movement and in the other part by an alteration of uncommon molecular mechanisms that are only partially described by this paper.

It will be important in future works to define more comprehensively the molecular pathway(s) involved and the specific RhoA cellular contributions. Clearing up signaling involved in the control of muscle regeneration, in particular, in response to muscle injury, may be essential to identify and design treatments for different pathological and traumatic conditions affecting skeletal muscle regeneration.

## Figures and Tables

**Figure 1 cells-12-02673-f001:**
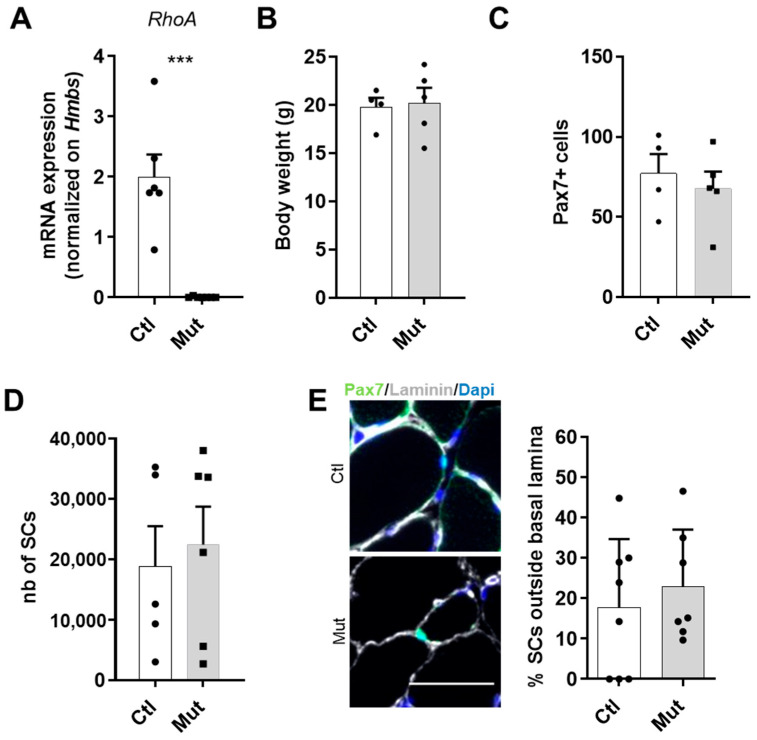
RhoA loss within SCs is not deleterious at basal state (**A**) *RhoA* mRNA expression by RT-qPCR in control and RhoA-deleted SCs. Data were normalized by *Hmbs* expression (n = 6). (**B**) Body weight of Tam injected or not mice (n = 4–5). (**C**) Absolute number of Pax7^+^ cells (n = 4–5). (**D**) SCs nGFP+ of Ctl and Mut muscles were isolated by FACS and counted (n = 5–6). (**E**) Representative images and quantification of SCs outside the basal lamina in control and RhoA-deleted *TA* muscle section immunostained for Pax7 (green), laminin (gray), and nuclear staining with DAPI (blue). Scale bar 50 μm. (**A**) Means ± SEM *** *p* < 0.0001 (Unpaired parametric *t*-test).

**Figure 2 cells-12-02673-f002:**
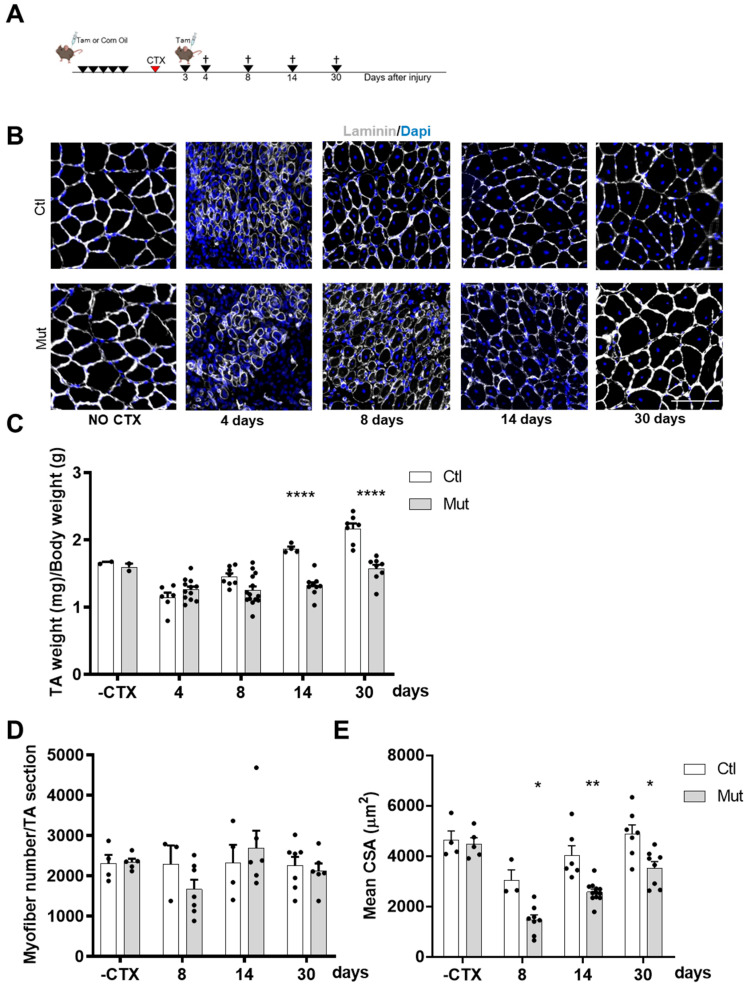
RhoA loss within SCs impairs skeletal muscle regeneration. (**A**) Transgenic mice were injected or not with Tam for five days in order to achieve RhoA deletion. Three days after CTX injection mice were injected with Tam. *TA* muscles were collected four, eight, fourteen, or thirty days after muscle damage. (**B**) *TA* sections were immunostained for laminin (gray) and nuclear staining with DAPI for Tam injected or not mice without CTX injection and four, eight, fourteen, and thirty days after CTX damage. Scale bar 100 µm. (**C**) The ratio of *TA* mass (mg) to body weight (g) before and four, eight, fourteen, and thirty days after CTX injection in Tam injected or not mice (n = 2–14). (**D**) Mean myofiber number before and four, eight, fourteen, and thirty days after CTX injection in Tam injected or not mice (n = 3–7). (**E**) Mean CSA (µm^2^) before and four, eight, fourteen, and thirty days after CTX injection in Tam injected or not mice (n = 3–12). (**C**,**E**) Means ± SEM * *p* < 0.05, ** *p* < 0.005, **** *p* < 0.0001 (ordinary two-way ANOVA with Sidak’s test).

**Figure 3 cells-12-02673-f003:**
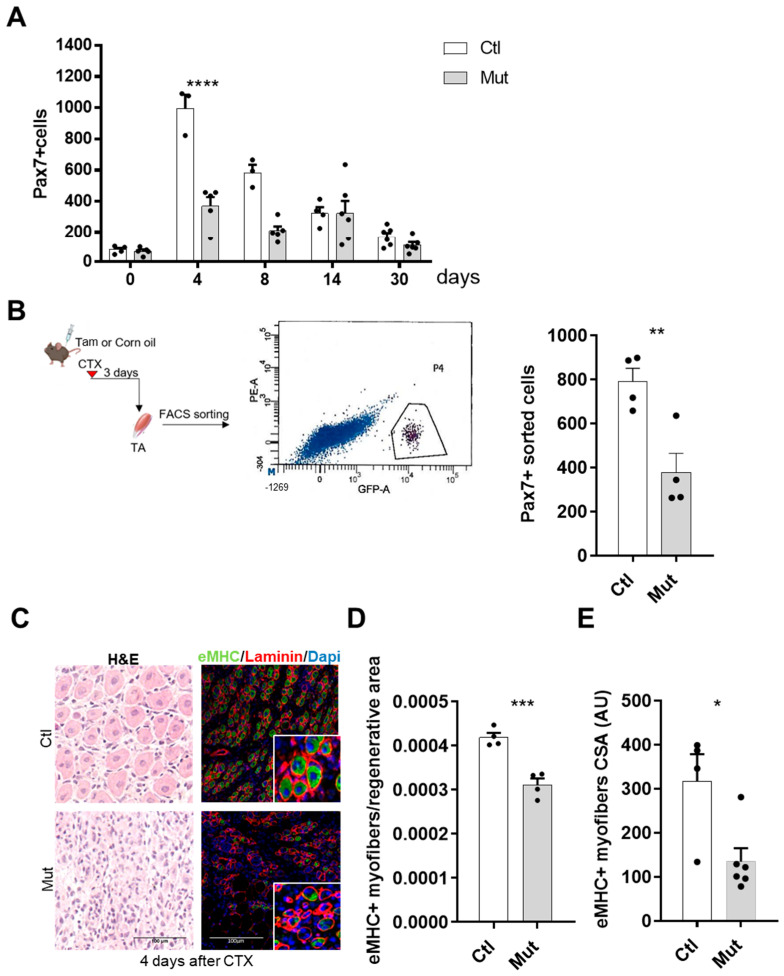
RhoA loss within SCs induces a delayed regeneration. (**A**) Absolute number of Pax7^+^ cells per muscle section (n = 3–6). (**B**) *TA* of Tam or not-injected mice were collected three days after CTX injection. SCs nGFP+ were isolated by FACS and counted. A representative image of the sorted cell population is shown according to their intrinsic size (FSC) and granularity (SSC) properties (exited by 488 nm laser). SCs nGFP+ of Ctl and Mut muscles were isolated by FACS and counted (n = 4). (**C**) Representative images of *TA* muscles of Tam injected or not mice four days after CTX injection H&E-stained or immunostained for eMHC (green), laminin (red), and nuclear staining with DAPI (blue). (**D**) Quantification of the number of fibers eMHC^+^ per area four days after CTX injection in Tam injected or not mice (n = 4). (**E**) Mean CSA (AU) of eMHC^+^ myofibers four days after CTX injection in Tam injected or not mice (n = 4–6). (**A**) Means ± SEM **** *p* < 0.0001 (ordinary two-way ANOVA with Sidak’s test). (**B**,**D**) Means ± SEM ** *p* < 0.01, *** *p* < 0.001 (Unpaired parametric *t*-test). (**E**) Means ± SEM * *p* < 0.01 (Mann Whitney test).

**Figure 4 cells-12-02673-f004:**
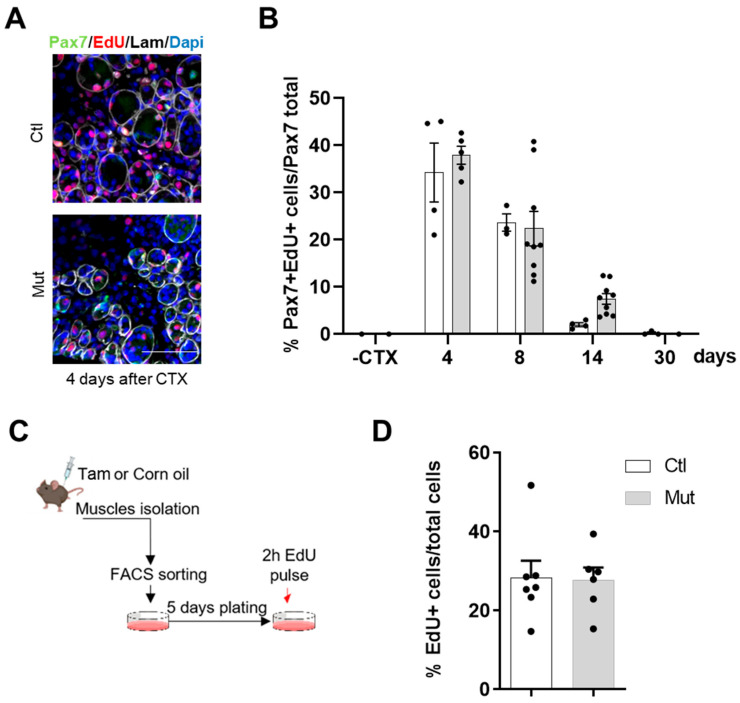
The absence of RhoA in SCs does not affect their proliferation. (**A**) Representative image of *TA* section of Tam injected or not mice immunostained for Pax7 (green), EdU (red), laminin (gray), and nuclear staining with DAPI four days after CTX injection. Scale bar 100 µm. (**B**) Percentage of Pax7^+^/EdU^+^ among Pax7^+^ cells in the *TA* section of Tam injected or not mice before and four, eight, fourteen, and thirty days after CTX injection (n = 3–9). (**C**) Hindlimb muscles of Tam injected or not mice were collected and SCs nPax7^+^ were isolated by FACS. Sorted cells were maintained in culture for five days and then submitted to EdU pulse. (**D**) Percentage of EdU^+^ cells in control and RhoA-deleted FACS-sorted SCs (n = 6–7).

**Figure 5 cells-12-02673-f005:**
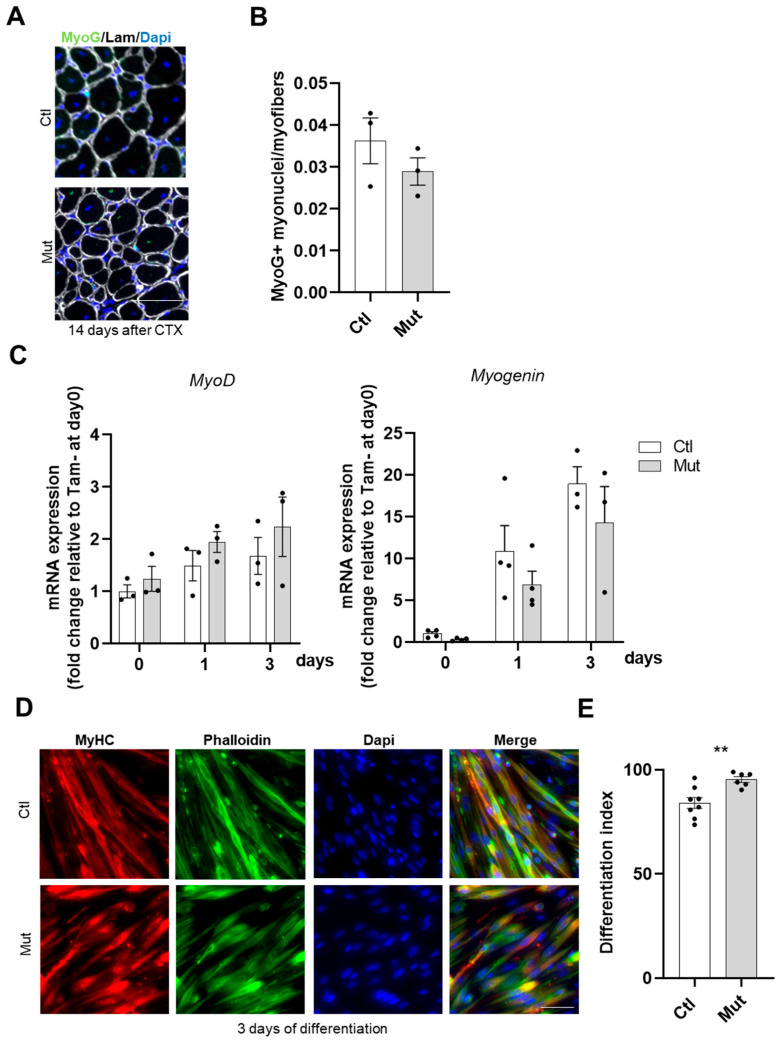
The absence of RhoA in SCs does not affect their differentiation. (**A**) Representative image of *TA* section of Tam injected or not mice fourteen days after CTX injection immunostained for Myogenin (green), laminin (gray), and nuclear staining with DAPI. Scale bar 100 µm. (**B**) Number of Myogenin^+^ (MyoG) nuclei per myofiber (n = 3). (**C**) Analysis of *MyoD* and *Myogenin* mRNA expression by RT-qPCR in FACS-sorted SCs control or RhoA-deleted cultured in rich medium (day 0) or 1 (day 1) and 3 (day 3) after differentiation. Data were normalized by *Hmbs* expression and relative to day 0 (n = 3–4). (**D**) Immunostaining for MyHC, nuclear staining with DAPI, and F-actin staining with phalloidin on FACS-sorted SCs control or RhoA-deleted three days after differentiation induction. Scale bar 50 µm. (**E**) Percentage of cells MyHC^+^ on total nuclear number in SCs control and RhoA-deleted three days after differentiation induction (n = 6–8). (**E**) Means ± SEM ** *p* < 0.005 (Unpaired parametric *t*-test).

**Figure 6 cells-12-02673-f006:**
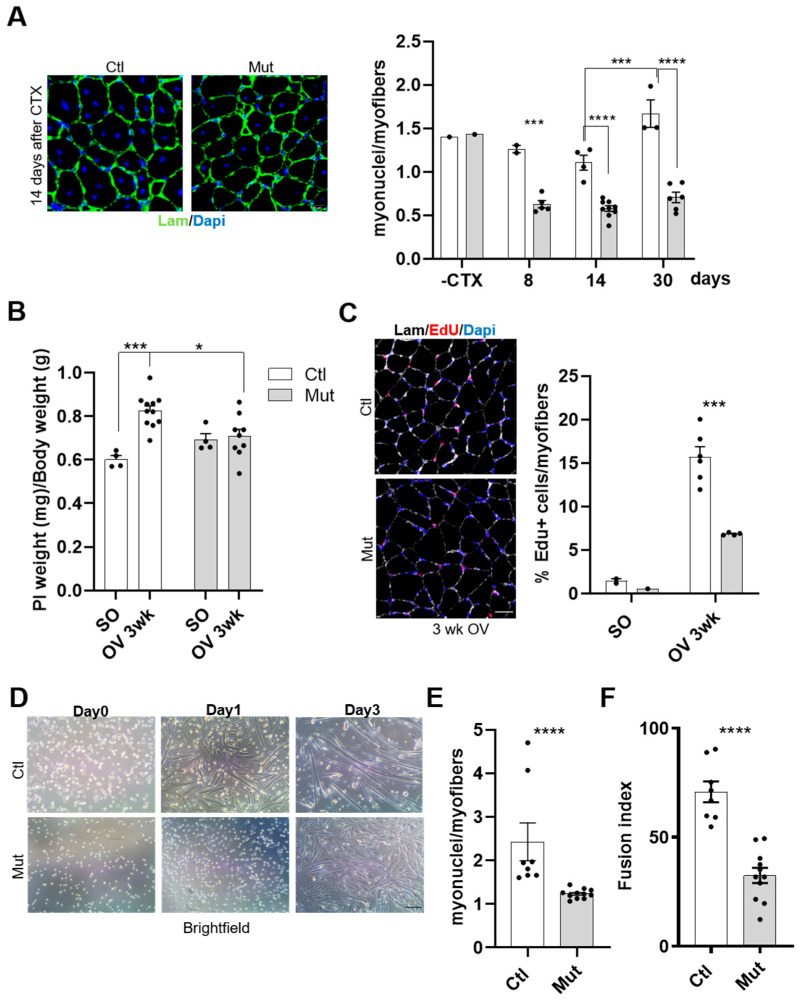
RhoA loss in SCs impairs their fusion in vivo and in vitro. (**A**) *TA* muscle section immunostained for laminin (green) and nuclear staining with DAPI for Tam injected or not mice fourteen days after CTX injection. Scale bar 100 µm. Number of nuclei (DAPI) within laminin^+^ sarcolemma per myofiber in *TA* muscles of Tam injected or not mice before, eight, fourteen, and thirty days after CTX injection (n = 2–9). (**B**) Ratio of *Plantaris* muscle mass (mg) to body weight (g) before (SO) and after three weeks of CH (OV 3 wk) in Tam or not injected mice (**C**) *Plantaris* muscle section immunostained for laminin (gray), EdU (red) and nuclear staining with DAPI for Ctl and Mut mice after three weeks of CH. Scale bar 100 µm. Percentage of EdU^+^ myofibers in *Plantaris* muscle section of Ctl and Mut mice before (SO) and after three weeks of CH (OV 3 w) (n = 2–6). (**D**) Phase-contrast representative images of FACS-sorted control and RhoA-deleted SCs cultured in rich medium (day 0) or one (day 1) and three (day 3) days after differentiation induction. Scale bar 100 µm. (**E**) Number of nuclei within MyHC^+^ myotubes (with more than two nuclei) in control and RhoA-deleted SCs induced to differentiate for three days (n = 8–11). (**F**) The proportion of nuclei within multinucleated cells in control and RhoA-deleted SCs three days after differentiation (n = 8–11). (**A**–**C**) Means ± SEM **** *p* < 0.0001, *** *p* < 0.0005 * *p* < 0.05 (ordinary two-way ANOVA with Sidak’s test). (**E**,**F**) Means ± SEM **** *p* < 0.0001 (Unpaired parametric *t*-test).

**Figure 7 cells-12-02673-f007:**
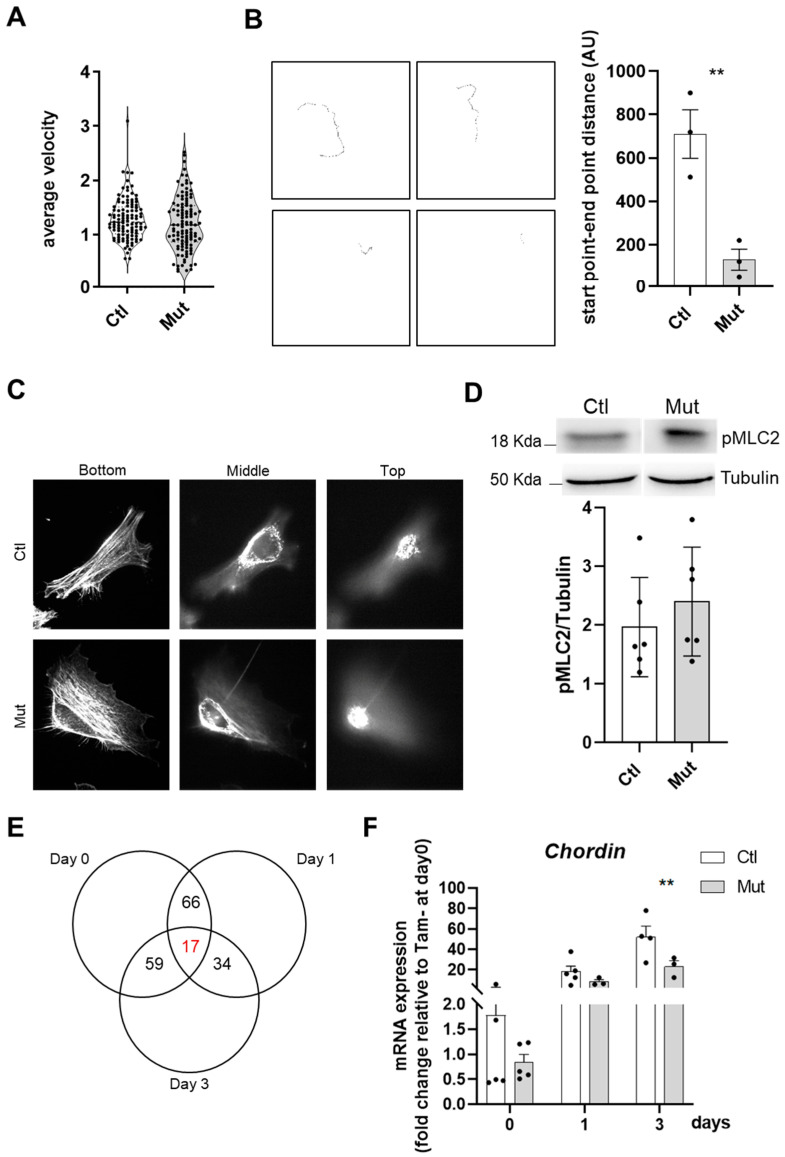
RhoA loss within SCs affects their motility but not their cytoskeleton. (**A**) Mean velocity (µm per minute) of control and RhoA-deleted myoblasts determined by time-lapse videomicroscopy. (**B**) Final images from representative video: individual cell traces used to calculate average cellular velocity. number of cells tracked per condition 109–112. Quantification of the distance between the start point and end point of single cells track. (**C**) Representative confocal projection of ȥ-section of F-actin staining (phalloidin) taken from the adherent (ventral) cell bottom and middle and proceeding up to the media-facing top of the control and RhoA-deleted myoblasts. (**D**) Representative western blotting image and quantification of pMCL2/Tubulin in control and RhoA-deleted myoblasts (three different cell cultures). (**E**) Venn diagram showing the intersection between genes differentially regulated by RhoA in myoblasts (Day 0), myocytes (Day 1), and myotubes (Day 3). In red is indicated the number of genes (17) that are modulated by RhoA independently of the differentiation muscle cells state. In the table, nine genes were identified by IPA whose expression is RhoA dependent and is deregulated in the same way in all the differentiation states. (**F**) Analysis of *Chordin* (*Chrd*) mRNA expression by RT-qPCR in FACS-sorted SCs control or RhoA-deleted, cultured in rich medium (Day 0) or 1 (Day 1) and 3 (Day 3) after differentiation. Data were normalized by *Hmbs* expression and relative to Tam- at Day 0 (n=2–6). (**B**) Means ± SEM ** *p* < 0.01 (Unpaired parametric *t*-test). (**F**) Means ± SEM ** *p* < 0.01, (ordinary two-way ANOVA with Sidak’s test).

## Data Availability

Data are contained within the article and Appendix A.

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
