# Peer review of "RhoA Is a Crucial Regulator of Myoblast Fusion"

_cells, 2023, doi:10.3390/cells12232673_

Round 1
Reviewer 1 Report
Comments and Suggestions for Authors
In the manuscript “RhoA is a crucial regulator of myoblast fusion”, C. Noviello and coworkers demonstrate that the small GTPase, RhoA has a role in myoblast-myoblast and myoblast-myofiber fusion, a crucial process for muscle formation during myogenesis and muscle regeneration. The authors used both in vitro and in vivo experimental models, including a tamoxifen-induced mouse model with satellite cell-specific RhoA deletion driven by the Pax7 promoter.
The manuscript is well written, the rationale is clear, the methods used are appropriate, and the conclusions sound right. The results are of sure interest in the myology field. However, some points should be addressed before publication.
Major points
1) Fig. 6A: the reduced numbers of myonuclei/myofibers in RhoA-depleted muscles can be due, at least in part, to the reduced numbers of Pax7+ cells available in these muscles, especially at day 8 (see Fig. 3A). This should be discussed.
2) Fig. 6C: the authors evaluated the percentages of EdU+ cells/myofiber in plantaris muscles of mutant and control mice in the absence or presence of compensatory hypertrophy protocol applied. Are the basal numbers of satellite cells/Plantaris muscle similar in mutant and control mice? Moreover, the percentages of double Pax7+/EdU+ cells (rather than EdU+ cells) should be evaluated to have a more reliable estimate of the proliferation extent of satellite cells in these conditions.
3) In the last part of the manuscript the authors investigated about modifications in the gene expression profile occurring in RhoA-deleted myoblasts during the differentiation process, in comparison with control myoblasts. By microarray analysis, they found several genes differentially regulated, with 17 genes displaying an altered expression in all the conditions tested (D0, D1 and D3). This part appears a bit confused and confusing for several reasons:
i) the altered expression of Adam12 is reported and discussed in the text; however, Adam20 appears as a modulated gene in the table of Fig. 7F, and Adam12 does not appear among the deregulated genes in the supplementary Excel file. Which is the Adam gene involved? Where Dnc results modulated?
ii) the “Relaxing signaling” gene category emerged as significantly deregulated in the absence of RhoA although with partially different implicated genes (i.e., Adcy8, Gnaz, Gng7, Nfkbib, Pde6d, Pde8b and Vegfa at day 1, and Apex1, Gnaz, Gng2, Gucy2d, Nfkbib, Pde6g and Vegfa at day 3). The authors focused just on Adcy8 gene. It could be interesting to expand the analysis to other deregulated genes.
iii) the authors performed RT-qPCR to evaluate the expression of decorin (Dnc) during the differentiation process in mutant and control myoblasts (Fig. S4). However, Dnc does not appear in any list of deregulated genes. What is the rationale of this analysis?
For the above reported reasons, I suggest to the authors to investigate deeply this part of the work or, alternatively, to remove it from the manuscript.
4) References should be extensively revised. Many references are reported in the text as reference numbers and other ones as “(Author et al.)”. Moreover, some references cited in the text are missing in the Reference list.
Minor points
1) Fig. 3A: the authors show the numbers of Pax7+ cells/muscle section at days 4-30 after cardiotoxin injury. The numbers of Pax7+ cells at day 0 should be also shown. This is important since in Fig. 1D the absolute numbers of Pax7+ cells in hindlimb muscles are reported, so that comparison with Fig. 3A is not possible. …
2) Fig. 7D: representative images of the Western blotting analysis should be shown.
3) In the text, tamoxifen is reported as “Tam” whereas in the figures it appears as “Tmx”. Please align the figure with the text.
4) Lines 387 and 406: Fig. S7 does not exist. It should be Fig. S4.
5) Fig. 3D: the number of eMHC-positive myofibers/regenerative area are reported. Please check the ordinate axis scale (0.0004 myofibers/area at maximum?).
6) Fig. 7F: gene symbols, which are referred to mouse, should be written in Italic and first letter upper case.
7) “MLC2” and “eMHC” should be defined.
8) Line 257: “Figure 5B” should be “Figure 5C”.
9) Line 263: “Figure 5C” should be “Figure 5D,E”.
10) Line 285: “Figure 6A-B” should be “Figure 6A”.
11) Line 289: “Figure 6B” should be “Figure 6A”.
12) Line 332 (Fig. 6 legend): “TA muscle” should be “Plantaris muscle”.
13) Legends for bars are missing in the graphs of Figures 2C,D, 3A, 4B, and 6A,B.
14) Fig. 3A: the current ordinate axis label is “Absolute nb of Pax7+ cells”; however, reported are the numbers of Pax7+ cells/muscle section. The term “absolute” should be removed.
Author Response
Thank you very much for taking the time to review this manuscript. We really appreciate the constructive comments and we are grateful for the improvement of the manuscript. Please find the detailed responses attached and the corresponding revisions/corrections highlighted/in track changes in the re-submitted files.

Reviewer 2 Report
Comments and Suggestions for Authors
The manuscript has been well described, and the reviewer has only minor comments.
Page 2, line 55, integrins have many families. Please specify which integrins the authors like to mention. Do M-cadherin and integrin localize in the same junction? Satellite cells exist below the basal membrane and interact with myofiber as well.
RhoA is a multifunctional regulatory protein. In page 16, from line 452, reconsider the description of the difference between ref 24 and this study to convince the readers.
Page 6, line 174, is it fine that TA is shown as Italic?
The sentence in 176-177 may be hard to understand for readers. Please revise the sentence [coming back in a normal range in the next day.]
In page 11, line 282, DAPi is frequently used.
Please consider before formal acceptance.
In page 17, line 515, [One gene] would be fine.
In page 20, the font size of sentences from line 614 seems larger.
In page 21, the font size of sentences from line 652 seems larger.
In page 23, the font size of sentences from line 720 seems larger.
In page 23, line 714, is Genom’ic fine?
Citation
In page 16 line 457 cite Brack et al paper.
In page 16, line 483, which references do the authors mean?
In page 20, line 567, is Lauriol et al cited?
In some references, the first author's initial is missing, such as refs 1,2,6,12,14, 15, 35, 41, and 43.
Author Response

(The authors gave the same response as above.)
